# Large-Size Suspended Mono-Layer Graphene Film Transfer Based on the Inverted Floating Method

**DOI:** 10.3390/mi12050525

**Published:** 2021-05-06

**Authors:** Qin Wang, Ying Liu, Fangsong Xu, Xiande Zheng, Guishan Wang, Yong Zhang, Jing Qiu, Guanjun Liu

**Affiliations:** College of Intelligence Science and Technology, National University of Defense Technology, Changsha 410073, China; wangqin14@126.com (Q.W.); liuying@nudt.edu.cn (Y.L.); fangsong2019@163.com (F.X.); persimmonkevin@outlook.com (X.Z.); guishan-wang@outlook.com (G.W.); zhangy21cn@126.com (Y.Z.)

**Keywords:** suspended graphene, IFM, damage mechanism, defects, stress concentration

## Abstract

Suspended graphene can perfectly present the excellent material properties of graphene, which has a good application prospect in graphene sensors. The existing suspended graphene pressure sensor has several problems that need to be solved, one of which is the fabrication of a suspended sample. It is still very difficult to obtain large-size suspended graphene films with a high integrity that are defect-free. Based on the simulation and analysis of the kinetic process of the traditional suspended graphene release process, a novel setup for large-size suspended graphene release was designed based on the inverted floating method (IFM). The success rate of the single-layer suspended graphene with a diameter of 200 μm transferred on a stainless-steel substrate was close to 50%, which is greatly improved compared with the traditional impregnation method. The effects of the defects and burrs around the substrate cavity on the stress concentration of graphene transfer explain why the transfer success rate of large-size suspended graphene is not high. This research lays the foundation for providing large-size suspended graphene films in the area of graphene high-precision sensors.

## 1. Introduction

As an emerging 2D material, graphene has superior material properties, including a high Young’s modulus, carrier mobility, light transmittance, and good piezoresistive effect [1,2,3,4,5,6]. In recent years, most components in graphene research have required substrate support; however, the substrate will participate in the transfer of graphene particles, which will result in an inability to exhibit the physical properties of graphene [7,8]. Isolated from the substrate, suspended graphene can almost perfectly preserve the pristine physical properties by combining the ultrahigh-vacuum environment and annealing clean process [9]. This merit provides suspended graphene with great promise in both fundamental physics and novel commercial applications [10,11,12,13,14]. Suspended graphene is also a perfect platform for nanoelectromechanical systems (NEMS), as ultra-sensitive sensors for mass, force, and light-emitting devices [15,16,17,18].

Recently, one of the main obstacles in its scientific research and commercial application has been the fabrication of a suspended sample. The mono-or-few-layer graphene is too thin to survive the fabrication processes, especially the release process from the solvent. Typical sizes of monolayer graphene membranes on a perforated substrate are in the order of tens of micrometers [19,20]. In order to obtain large-size suspended graphene films, researchers are continuously improving suspension release methods [21,22,23]. Lee’s improved inverted floating method (IFM) obtained the largest single-layer suspended graphene film, with a diameter of up to 500 μm [24].

Based on the simulation and analysis of the dynamic process of the traditional release process, this paper designed a new scheme to transfer the single-layer graphene film on a stainless-steel substrate based on the IFM method. The stress concentration caused by burrs at the edge of cavities was simulated and analyzed, which would be responsible for the low yield of the suspended graphene with a size beyond 500 μm. Our efforts on large-size suspended graphene sample fabrication will promote applications of graphene-based novel sensors and electronics.

## 2. Analysis of Graphene Film Damage Mechanism

### 2.1. Analysis of Damage Mechanism during Suspension Release

The widely used method for producing high-quality graphene is chemical vapor deposition (CVD). According to the operating environment of CVD graphene film, there are two transfer methods commonly used, namely: the dry transfer method and the wet transfer method. The dry transfer method is usually favored for small-batch sample preparation at a lab level. In this paper, we focus on the wet transfer method to help solve the graphene fracture problem.

The last step of the wet transfer of suspended graphene is the release process. After removing the polymethyl methacrylate (PMMA) layer from the graphene membranes using acetone, the samples were removed from the acetone and were gradually dried under atmospheric conditions. Acetone on the top surface of the graphene evaporated faster than acetone inside the hole beneath the graphene because of the tiny volume of the hole in the perforated substrate. The acetone remaining inside the hole exerted a mechanical load on the graphene. The main reason for the rupture of the graphene film is the sudden stress during the suspension release process, which makes the film stress overload and rupture. Figure 1b shows the schematic diagram of the film rupture caused by the solution surface tension. Once all the liquid under the film is dry and the film has been loaded by capillary pressure and the three-phase interface, the suspension of the graphene film is achieved, as shown in Figure 1a.

When the bottom of the graphene film is completely covered by liquid, as shown in Figure 2a, the hole can be modeled as an interface containing droplets so as to evaluate the load caused by the surface tension [25]:(1)$$P_{cap}=\frac{4\lambda \sin\;\theta }{D}$$
where $P_{cap}$ is the capillary pressure, λ is the surface tension of the liquid, θ is the contact angle between the capillary wall and the liquid, and *D* is the diameter of the capillary.

It is known from Equation (1) that the smaller the surface tension of the solvent, the smaller the contact angle between the film and the liquid, and the smaller the load when the diameter of the film is constant.

Using COMSOL software to model the process of this stage, the graphene elastic modulus is 955 GPa, the Poisson’s ratio is 0.165, the graphene film thickness is 0.335 nm, and the water surface tension is 0.0728 N/m [26]. When the bottom surface of the graphene film is covered by water, the load applied on the graphene film is about 4155 Pa, according to Equation (1), and the result is shown in Figure 2a.

As the liquid evaporates, the central area of the bottom surface of the graphene is exposed to air, and a new three-phase interface composed of graphene, liquid, and air is formed, as shown in Figure 2b. The extra load generated by the three-phase interface is as follows:(2)$$F_{T}=\pi D_{T}\gamma \sin\theta$$

Initially, the diameter $D_{T}$ is very small. As the liquid evaporates, the diameter of the three-phase interface gradually increases. Both the theoretical analysis and simulation reveal that the extra load leads to a large deflection of the graphene membrane, which subsequently results in a huge stress at the centre regime. Furthermore, the stress concentrates around the unavoidable pin-holes on the CVD graphene surface. This is one of the main causes of graphene damage in traditional methods.

### 2.2. Stress Concentration Analysis of Thin Film Defects

Observing the micro-defects of the film through a microscope, it is found that the micro-defects in the suspended graphene fracture are mainly elliptical holes and round holes. COMSOL has been used to model and simulate the stress concentration caused by the micro-defects of the graphene film. Considering that the film thickness is much smaller than the film diameter, it is simplified to a plane stress problem. The stress concentration model is shown in Figure 3.

Considering that the defects are much smaller than the size of the film, the stress concentration coefficients of the small holes are similar in each position of the film, which can be simplified in the software to open the center circular hole and the elliptical hole film stress concentration modeling analysis. Based on the symmetry of structure and load, only 1/4 of the model is needed for the analysis. The film diameter *D* = 70 μm and the boundary load is taken as the maximum stress during the release process, *q* = 8 GPa and the radius of the hole *a* = 1 μm. Figure 4 shows that as the stress concentration simulation result of the open hole film, the stress reaches 16GPa.

For the stress concentration analysis of the elliptical hole film, the b/a value is taken as 0.2. The simulation result is shown in Figure 5. The concentrated stress reaches 78.1 GPa, reaching half of the fracture strength of the graphene film with a grain boundary structure [27]. It can be seen that the stress concentration in the circular pinhole is significantly lower than the stress concentration obtained from the elliptical pinhole, and the graphene film with the elliptical pinhole is more likely to break. The number of defects increases in proportion to the area of the film, which explains why large-size graphene films are more likely to break.

Reducing *F_T_* or avoiding the formation of a three-phase interface is key to improving the success rate of graphene suspension. For this reason, this paper carries out the following comparative tests: the (1) conventional dipping method, (2) low-stress solution dipping method, and (3) IFM method.

## 3. Experimental Section

CVD graphene was used in the experiment, and it was transferred to a stainless-steel substrate with the thickness of 3 mm. The stainless-steel substrate was laser-drilled with a diameter ranging from 10 µm to 1500 µm. The process of removing PMMA with acetone and then drying is the most important part of the entire transfer process. Most of the graphene films are broken in this step. During the experiment, the experimental conditions, such as the quality of CVD graphene, etching of the copper foil, and handling of the graphene during transfer (Appendix A), were kept consistent. The graphene used in the three sets of experiments was all cut on a piece of 10 × 10 cm copper-based graphene.

As shown in Figure 6, method 1 is a conventional dipping method. The sample was directly immersed in an acetone solution. After the PMMA was removed by the acetone, the sample was taken out and dried. In method 2, we first removed the PMMA in an acetone solution, then transferred the sample to low-tension methoxy nonafluorobutane (C_4_F_9_OCH_3_) and replaced the acetone with C_4_F_9_OCH_3_. As C_4_F_9_OCH_3_ cannot dissolve the PMMA layer, it was just used for reducing the surface tension of the acetone. However, in method 2, graphene needed to be transferred from the acetone solution to the C_4_F_9_OCH_3_ solution in the actual experiment process. Other factors such as the influence of liquid disturbance and unstable operation of the transfer process affected the transfer process. As a result, the transfer success rate of method 2 did not improve much compared with that of method 1.

In order to avoid the formation of a three-phase interface and to reduce the stress, a novel setup for large-size suspended graphene release was designed based on the IFM method. In this way, the sample was rinsed without being immersed in liquid, but rather letting the sample float on the surface of an acetone bath. The concept schematic diagram is depicted in Figure 7. The substrate covered with PMMA was faced down towards the floating acetone solvent in the trench. After the removal of the PMMA, a low-tension solvent, C_4_F_9_OCH_3_ solvent, was slowly injected so as to substitute the acetone. Finally, we removed the suspended graphene sample and dried it with a weak nitrogen flow. The setup was home-made by a 3D printer. The dark part in the figure is the acetone container with a channel. The silvery white part is the stainless-steel substrate for the graphene transfer. There was no need to move the sample during the experiment, which reduced the difficulty of the operation process. The IFM method prevented the acetone liquid from entering the holes above the graphene film and avoided the formation of the three-phase interface, which improved the transfer success rate of the large-size suspended graphene film.

## 4. Characterization and Analysis

### 4.1. SEM Analysis

When analyzing the suspended graphene film obtained by the three methods by scanning electron microscopy, it was found that the success rate of conventional method 1 was extremely low. Figure 8 shows the SEM images of the suspended graphene films obtained by method 1 and method 3. Figure 9 shows the transfer success rate of the three methods (Appendix A). The success rate was defined as the ratio of the number of intact graphene films before and after PMMA was removed. Method 1 has extremely low graphene coverage, and method 2 has a greater success rate than method 1, which indicates that the surface tension of the liquid was the main cause of damage. The transfer success rate of 50 μm holes in the two methods was less than 40%. When using inverted float transfer, it could be seen that most of the holes were completely covered. As the transfer size increased, the coverage rate also decreased, but the success rate was greatly improved compared with conventional method 1 and method 2. The success rate of the 200 μm suspended graphene was close to 50%. The comparision of three release methods showed that the liquid in the substrate hole during the drying process was one of the main reasons for the fracture of graphene. The IFM method successfully avoided forming the three-phase interface and achieved the best performance.

### 4.2. Raman Test and Analysis

Figure 10 shows the Raman test results of a 200 μm single-layer suspended graphene film successfully transferred. The Raman spectrum of most areas of suspended graphene is shown in Figure 10c. According to the 2D peak shape of the spectrum and the I_2D_/I_G_ ratio, it can be seen that the CVD graphene used in this report mostly consisted of monolayer graphene [28,29,30]. The Raman spectrum of part of the bright spot area is the same as in Figure 10d. It can be seen that there is multilayer graphene in the CVD graphene. From the Raman spectrum, it can be seen that there was a D peak on the graphene Raman spectrum, which indicates that the CVD graphene had defects or impurities [31]. The analysis reasons are as follows: the inevitable lattice defects during the growth of CVD graphene, the influence of PMMA residue after transfer, and the influence of the adsorbent in the air.

It can be seen from Figure 9 that the transfer success rate of the graphene film with a size of 500 μm was dramatically reduced. The yield was close to zero, which was far lower than what we predicted. According to the SEM image of the suspended graphene obtained by the experiment, the burrs at the edge of the hole caused by laser drilling may be the cause of the film rupture and the low yield.

### 4.3. Simulation of Spike Stress Concentration

The optical and SEM observations show that there were many sharp burrs on the edge of the cavity. During the transfer process, the sample was inevitably immersed in the acetone solution. When the sample was immersed in the acetone solution for 1 mm, the density of acetone at room temperature was 0.000788 g/cm^3^. We could then calculate the surface pressure of about 1Pa on the graphene surface by P=ρgh. Using COMSOL to simulate the stress concentration of the film edge burr, the results are shown in Figure 11. The graphene film stress at the edge burr was as high as 80 GPa. Moreover, the film stress was proportional to the sinking depth. As the sinking depth increased, the stress became more serious. This implies that the edge quality of the hole played an unexpected but important role in the production of suspended graphene, and the influence may have exceeded the three-phase interface.

## 5. Conclusions

In conclusion, the cracking of graphene in the conventional wet method generally occurs during the formation of the three-phase interface. Through a simulation analysis of the stress concentration effect of CVD graphene film defects, the graphene film with elliptical defects is easier to crack than the graphene with circular defects. The number of defects increases in proportion to the film area, which explains why large-size graphene films are more likely to break. Based on the understanding of the damage mechanism, an inverted floating transfer experiment was designed to prevent the acetone from entering the through holes, and we replaced the acetone with C_4_F_9_OCH_3_ solvent when transferring a single-layer graphene film. The success rate of the single-layer suspended graphene with a diameter of 200 μm transferred on a stainless-steel substrate was close to 50%, which is greatly improved compared with the traditional method. From the simulation, we found that that the edge quality of the hole also had an important influence on the success rate of the suspended graphene transfer. This paper lays the foundation for the study of sensors and electronics based on the suspended graphene film.

## Figures and Tables

**Figure 1 micromachines-12-00525-f001:**
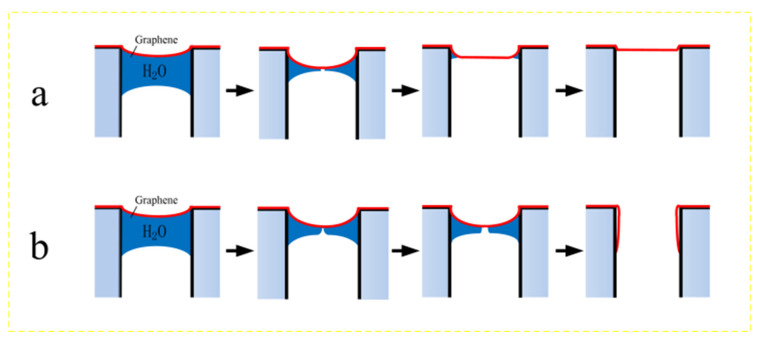
The schematic diagram of the suspension release process. (**a**) The process of obtaining complete suspended graphene film after drying. (**b**) The sudden stress of the solution release process causes the graphene film to rupture.

**Figure 2 micromachines-12-00525-f002:**
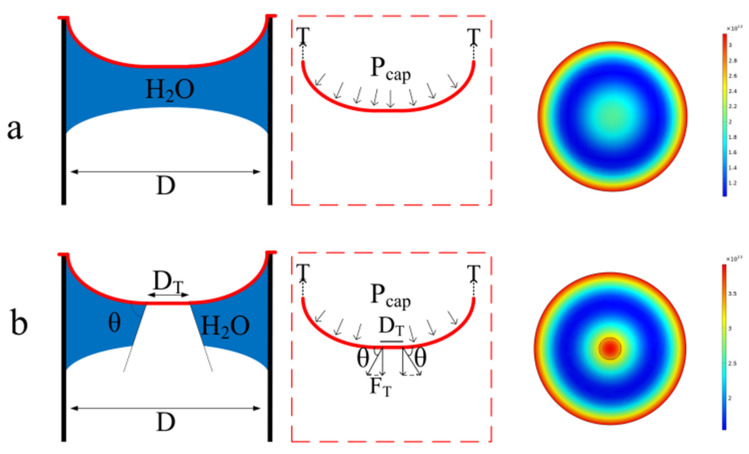
Two-stage force diagram and simulation diagram. (**a**) The bottom surface of graphene is completely covered by liquid. (**b**) Three-phase interface formation stage.

**Figure 3 micromachines-12-00525-f003:**
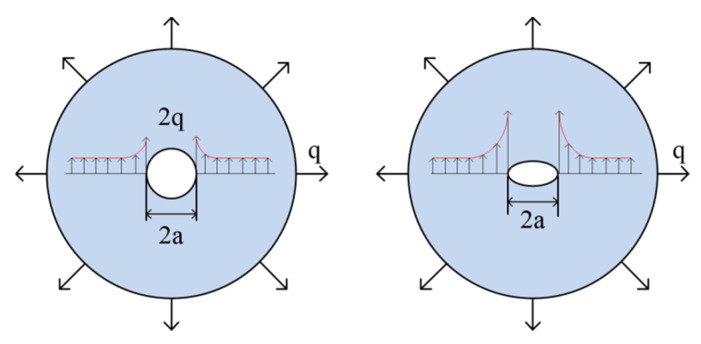
The mechanics model of the film with round and elliptical holes.

**Figure 4 micromachines-12-00525-f004:**
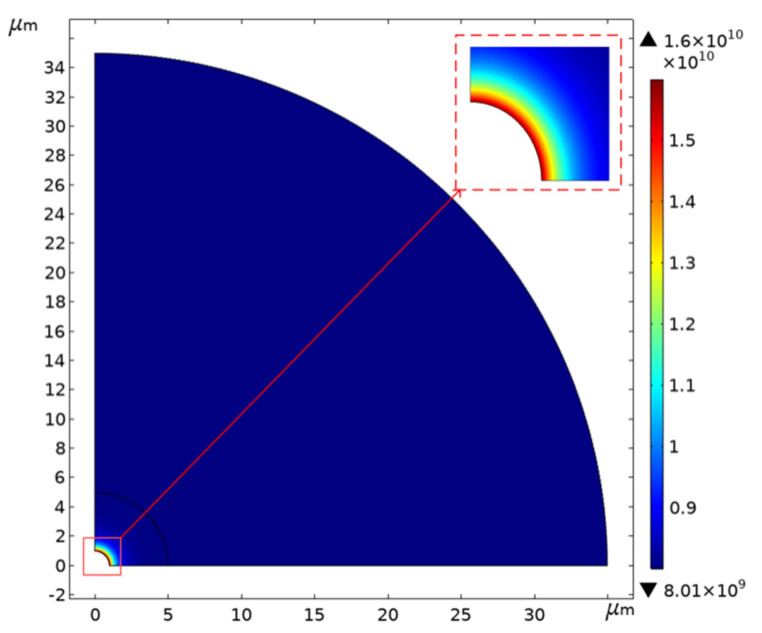
The simulation results of the stress distribution around the circular hole.

**Figure 5 micromachines-12-00525-f005:**
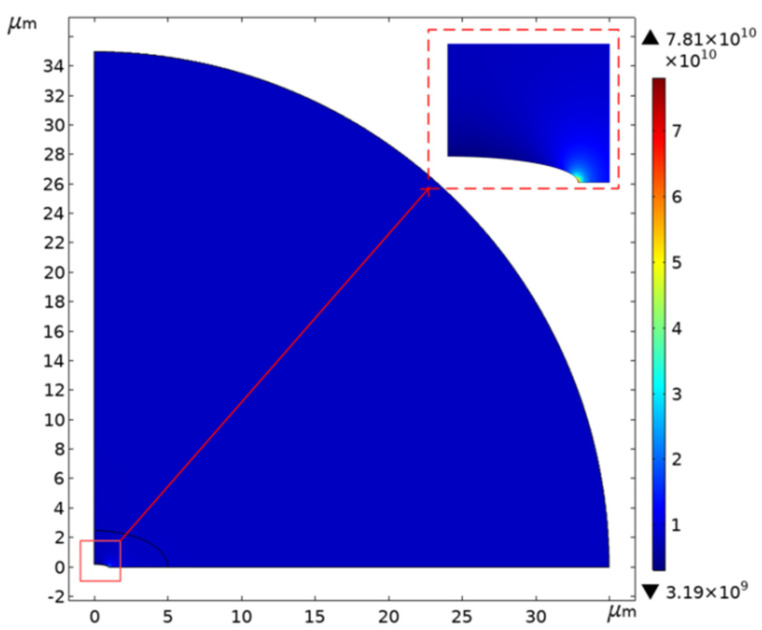
The simulation results of stress distribution around the elliptical hole.

**Figure 6 micromachines-12-00525-f006:**
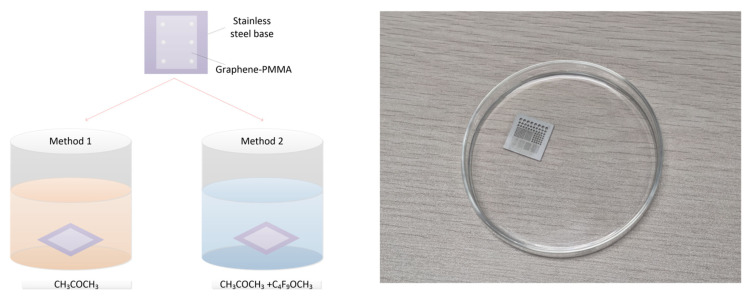
The schematic diagram of two conventional dipping methods to remove the PMMA.

**Figure 7 micromachines-12-00525-f007:**
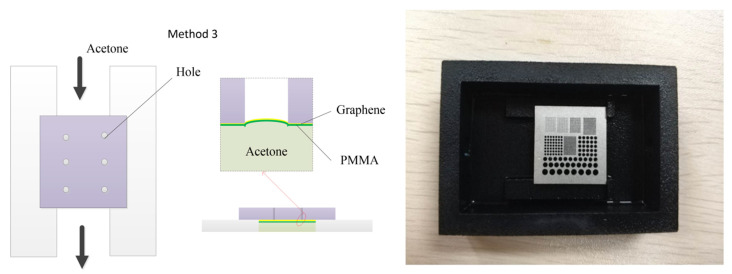
The schematic diagram of the PMMA removal by the inverted floating method (IFM) method.

**Figure 8 micromachines-12-00525-f008:**
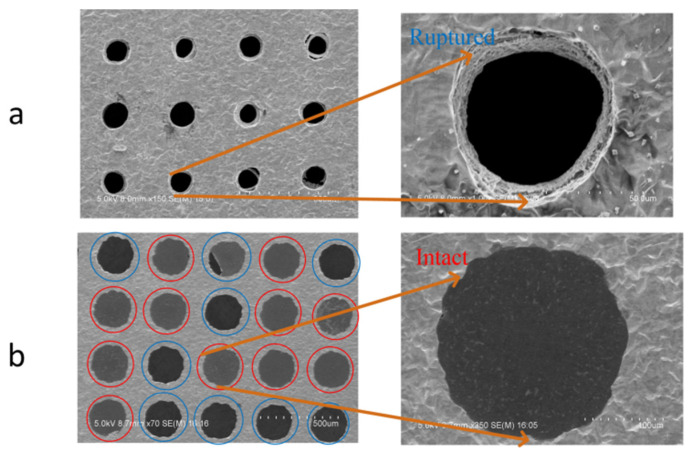
The SEM image of method 1 and method 3 after removing the PMMA. (**a**) The image of 80 μm suspended graphene transferred by method 1. (**b**) The image of 200 μm suspended graphene transferred by method 3.

**Figure 9 micromachines-12-00525-f009:**
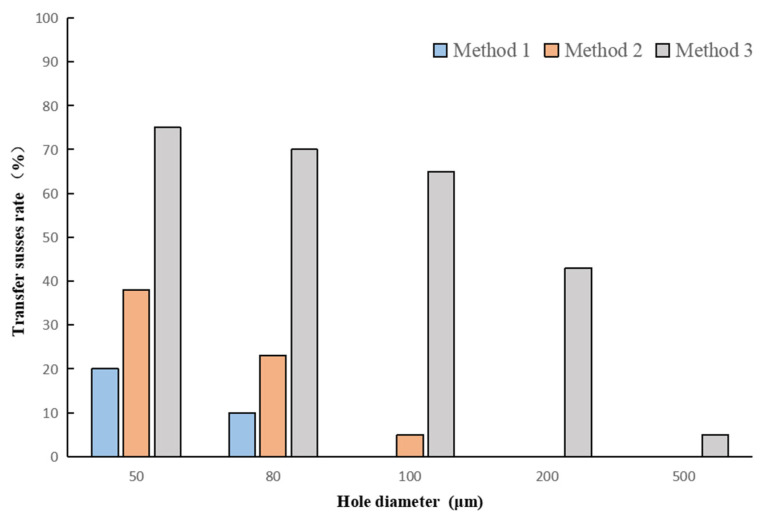
The statistics of the success rate for the three methods.

**Figure 10 micromachines-12-00525-f010:**
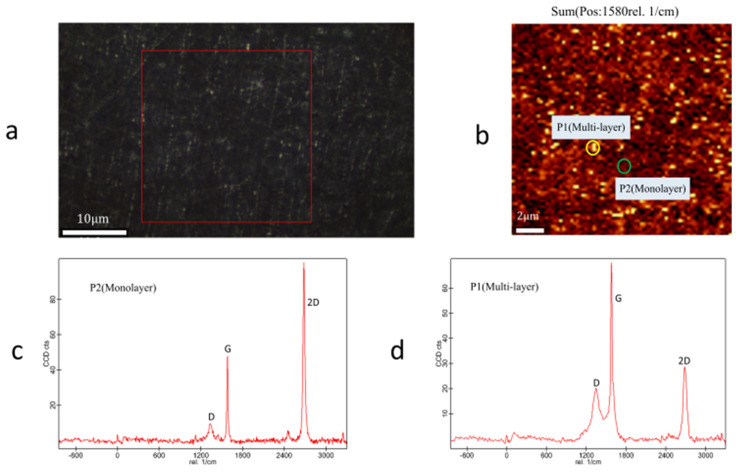
The suspended graphene Raman test. (**a**) Using a 532nm laser to excite the suspended graphene region. (**b**) Raman spectrum in the red frame region generates an image at 1580rel .1/cm. (**c**) Raman spectrum at P2. (**d**) Raman spectrum at P1.

**Figure 11 micromachines-12-00525-f011:**
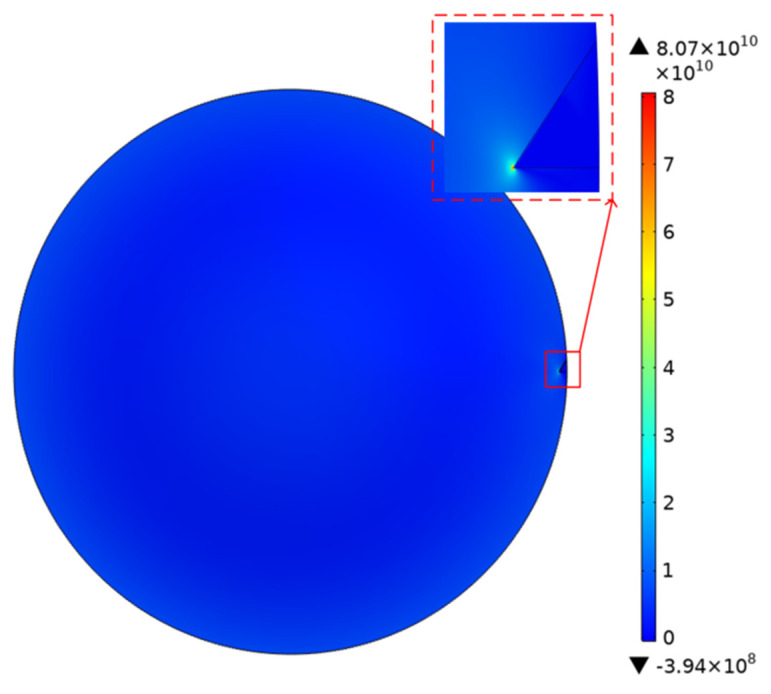
The simulation results of the film stress distribution when there are spikes on the edge of the substrate.

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
