# Peer review of "Large-Size Suspended Mono-Layer Graphene Film Transfer Based on the Inverted Floating Method"

_micromachines, 2021, doi:10.3390/mi12050525_

Round 1

Reviewer 1 Report

In this paper, Authors reported a unique system to achieve the suspended graphene over large area and well described the simulation result with proper experimental evidences. So that Reviewer recommend to publish the manuscript in this Micromachines, once after they address a couple of questionable point as below.

  1. In page 2 line 58-64, it is hard to understand the explanation how two system, ruptured and successful suspension could be defined.
  2. In page 5 line 132. The quality and reproducibility of CVD graphene is critical for the process, how did Author confirm the consistent quality of CVD graphene as said 'were kept consistent' ?
  3. In page 6 line  152. How did Author determine the PMMA layer is completely removed out by acetone ?
  4. In Fig. 8, why did you use the different size of hole substrate, even you intended to see the direct comparison between two method.
  5. Regarding Raman test and analysis Authors provided, how much difference in Raman spectrum (in terms of quality) before and after suspension process in the case of IFM method? 

Reviewer 2 Report

The paper describes a method for the fabrication of suspended graphene films. The experimental procedure is not correctly described (characterization methods, synthesis methods)... the authors have not provided all the details about the experiments. Important calculation parameters as the "success rate" must be adequate described. The correlation between experimental and theoretical calculations has to be also proved.

Round 2

Reviewer 2 Report

I am satisfied with the changes made to the manuscript and I would recommend its publication.